# On the Expressivity of Markov Reward

**David Abel**
DeepMind
dmabel@deepmind.com

**Will Dabney**
DeepMind
wdabney@deepmind.com

**Anna Harutyunyan**
DeepMind
harutyunyan@deepmind.com

**Mark K. Ho**
Department of Computer Science
Princeton University
mho@princeton.edu

**Michael L. Littman**
Department of Computer Science
Brown University
mlittman@cs.brown.edu

**Doina Precup**
DeepMind
doinap@deepmind.com

**Satinder Singh**
DeepMind
baveja@deepmind.com

## Abstract

Reward is the driving force for reinforcement-learning agents. This paper is dedicated to understanding the expressivity of reward as a way to capture tasks that we would want an agent to perform. We frame this study around three new abstract notions of "task" that might be desirable: (1) a set of acceptable behaviors, (2) a partial ordering over behaviors, or (3) a partial ordering over trajectories. Our main results prove that while reward can express many of these tasks, there exist instances of each task type that no Markov reward function can capture. We then provide a set of polynomial-time algorithms that construct a Markov reward function that allows an agent to optimize tasks of each of these three types, and correctly determine when no such reward function exists. We conclude with an empirical study that corroborates and illustrates our theoretical findings.

## 1 Introduction

How are we to use algorithms for reinforcement learning (RL) to solve problems of relevance in the world? Reward plays a significant role as a general purpose signal: For any desired behavior, task, or other characteristic of agency, there must exist a reward signal that can incentivize an agent to learn to realize these desires. Indeed, the expressivity of reward is taken as a backdrop assumption that frames RL, sometimes called the reward hypothesis: "...all of what we mean by goals and purposes can be well thought of as maximization of the expected value of the cumulative sum of a received scalar signal (reward)" [53, 29, 6]. In this paper, we establish first steps toward a systematic study of the reward hypothesis by examining the expressivity of reward as a signal. We proceed in three steps.

**1. An Account of "Task".** As rewards encode tasks, goals, or desires, we first ask, "what *is* a task?". We frame our study around a thought experiment (Figure 1) involving the interactions between a designer, Alice, and a learning agent, Bob, drawing inspiration from Ackley and Littman [2], Sorg [50], and Singh et al. [46]. In this thought experiment, we draw a distinction between how Alice thinks of a task (TASKQ) and the means by which Alice incentivizes Bob to pursue this task (EXPRESSIONQ). This distinction allows us to analyze the expressivity of reward as an answer to the latter question, conditioned on how we answer the former. Concretely, we study three answers to the TASKQ in the context of finite Markov Decision Processes (MDPs): A task is either (1) a set of

35th Conference on Neural Information Processing Systems (NeurIPS 2021).

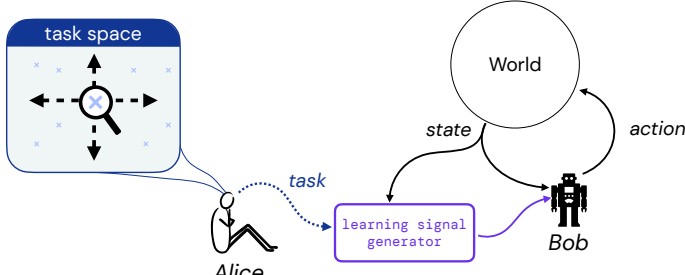

Figure 1: Alice, Bob, and the artifacts of task definition (blue) and task expression (purple).

acceptable behaviors (policies), (2) a partial ordering over behaviors, or (3) a partial ordering over trajectories. Further detail and motivation for these task types is provided in Section 3, but broadly they can be viewed as generalizations of typical notions of task such as a choice of goal or optimal behavior. Given these three answers to the TASKQ, we then examine the *expressivity* of reward.

**2. Expressivity of Markov Reward.**   The core of our study asks whether there are tasks Alice would like to convey—as captured by the answers to the TASKQ—that admit no characterization in terms of a Markov reward function. Our emphasis on Markov reward functions, as opposed to arbitrary history-based reward functions, is motivated by several factors. First, disciplines such as computer science, psychology, biology, and economics typically rely on a notion of reward as a numerical proxy for the *immediate* worth of states of affairs (such as the financial cost of buying a solar panel or the fitness benefits of a phenotype). Given an appropriate way to describe states of affairs, Markov reward functions can represent immediate worth in an intuitive manner that also allows for reasoning about combinations, sequences, or re-occurrences of such states of affairs. Second, it is not clear that general history-based rewards are a reasonable target for learning as they suffer from the curse of dimensionality in the length of the history. Lastly, Markov reward functions are the standard in RL. A rigorous analysis of which tasks they can and cannot convey may provide guidance into when it is necessary to draw on alternative formulations of a problem. Given our focus on Markov rewards, we treat a reward function as accurately *expressing* a task just when the value function it induces in an environment adheres to the constraints of a given task.

**3. Main Results.**   We find that, for all three task types, there are environment–task pairs for which there is no Markov reward function that realizes the task (Theorem 4.1). In light of this finding, we design polynomial-time algorithms that can determine, for any given task and environment, whether a reward function exists in the environment that captures the task (Theorem 4.3). When such a reward function does exist, the algorithms also return it. Finally, we conduct simple experiments with these procedures to provide empirical insight into the expressivity of reward (Section 5).

Collectively, our results demonstrate that there are tasks that cannot be expressed by Markov reward in a rigorous sense, but we can efficiently construct such reward functions when they do exist (and determine when they do not). We take these findings to shed light on the nature of reward maximization as a principle, and highlight many pathways for further investigation.

## 2   Background

RL defines the problem facing an agent that learns to improve its behavior over time by interacting with its environment. We make the typical assumption that the RL problem is well modeled by an agent interacting with a finite Markov Decision Process (MDP), defined by the tuple $(\mathcal{S}, \mathcal{A}, R, T, \gamma, s_0)$. An MDP gives rise to deterministic behavioral policies, $\pi : \mathcal{S} \to \mathcal{A}$, and the value, $V^\pi : \mathcal{S} \to \mathbb{R}$, and action–value, $Q^\pi : \mathcal{S} \times \mathcal{A} \to \mathbb{R}$, functions that measure their quality. We will refer to a Controlled Markov Process (CMP) as an MDP without a reward function, which we denote $E$ for environment. We assume that all reward functions are deterministic, and may be a function of either state, state-action pairs, or state-action-state triples, but *not* history. Henceforth, we simply use "reward function" to refer to a deterministic Markov reward function for brevity, but note that more sophisticated settings beyond MDPs and deterministic Markov reward functions are important directions for future work. For more on MDPs or RL, see the books by Puterman [41] and Sutton and Barto [54] respectively.

## 2.1 Other Perspectives on Reward

We here briefly summarize relevant literature that provides distinct perspectives on reward.

**Two Roles of Reward.** As Sorg [50] identifies (Chapter 2), reward can both define the task the agent learns to solve, and define the "bread crumbs" that allow agents to efficiently learn to solve the task. This distinction has been raised elsewhere [2, 46, 47], and is similar to the extrinsic-intrinsic reward divide [45, 66]. Tools such as reward design [34, 51] or reward shaping [36] focus on offering more efficient learning in a variety of environments, so as to avoid issues of sparsity and long-term credit assignment. We concentrate primarily on reward's capacity to express a *task*, and defer learning dynamics to an (important) stage of future work.

**Discounts, Expectations, and Rationality.** Another important facet of reward is how it is used in producing behavior. The classical view offered by the Bellman equation (and the reward hypothesis) is that the quantity of interest to maximize is expected, discounted, cumulative reward. Yet it is possible to disentangle reward from the expectation [5], to attend only to ordinal [60] or maximal rewards [26], or to adopt different forms of discounting [61, 11]. In this work, we take the standard view that agents will seek to maximize *value* for a particular discount factor $\gamma$, but recognize that there are interesting directions beyond these commitments, such as inspecting the limits of reward in constrained MDPs as studied by Szepesvári [56]. We also note the particular importance of work by Pitis [40], who examines the relationship between classical decision theory [59] and MDPs by incorporating additional axioms that account for stochastic processes with discounting [24, 35, 48, 49]. Drawing inspiration from Pitis [40] and Sunehag and Hutter [52], we foresee valuable pathways for future work that further makes contact between RL and various axioms of rationality.

**Preferences.** In place of numerical rewards, preferences of different kinds may be used to evaluate an agent's behaviors, drawing from the literature on preference-learning [25] and ordinal dynamic programming [8, 35, 48]. This premise gives rise to *preference-based reinforcement learning* (PbRL) in which an agent interacts with a CMP and receives evaluative signals in the form of preferences over states, actions, or trajectories. This kind of feedback inspires and closely parallels the task types we propose in this work. A comprehensive survey of PbRL by Wirth et al. [64] identifies critical differences in this setup from traditional RL, categorizes recent algorithmic approaches, and highlights important open questions. Recent work focuses on analysing the sample efficiency of such methods [65, 38] with close connections to learning from human feedback in real time [23, 32, 7].

**Teaching and Inverse RL.** The inverse RL (IRL) and apprenticeship learning literature examine the problem of learning directly from behavior [37, 1]. The classical problem of IRL is to identify which reward function (often up to an equivalence class) a given demonstrator is optimizing. We emphasize the relevance of two approaches: First, work by Syed et al. [55], who first illustrate the applicability of linear programming [22] to apprenticeship learning; and second, work by Amin et al. [4], who examine the *repeated* form of IRL. The methods of IRL have recently been expanded to include variations of *cooperative* IRL [14], and *assistive* learning [43], which offer different perspectives on how to frame interactive learning problems.

**Reward Misspecification.** Reward is also notoriously hard to specify. As pointed out by Littman et al. [30], "putting a meaningful dollar figure on scuffing a wall or dropping a clean fork is challenging." Along these lines, Hadfield-Menell et al. [16] identify cases in which well-intentioned designers create reward functions that produce unintended behavior [39]. MacGlashan et al. [33] find that human-provided rewards tend to depend on a learning agent's entire policy, rather than just the current state. Further, work by Hadfield-Menell et al. [15] and Kumar et al. [27] suggest that there are problems with reward as a learning mechanism due to misspecification and reward tampering [10]. These problems have given rise to approaches to *reward learning*, in which a reward function is inferred from some evidence such as behavior or comparisons thereof [20].

**Other Notions of Task.** As a final note, we highlight alternative approaches to task specification. Building on the Free Energy Principle [13, 12], Hafner et al. [17] consider a variety of task types in terms of minimization of distance to a desired target distribution [3]. Alternatively, Littman et al. [30] and Li et al. [28] propose variations of *linear temporal logic* (LTL) as a mechanism for specifying a task to RL agents, with related literature extending LTL to the multi-task [58] and multi-agent

[18] settings, or using reward machines for capturing task structure [19]. Jothimurugan et al. [21] take a similar approach and propose a task specification language for RL based on logical formulas that evaluate whether trajectories satisfy the task, similar in spirit to the logical task compositions framework developed by Tasse et al. [57]. Many of these notions of task are more general than those we consider. A natural direction for future work broadens our analysis to include these kinds of task.

# 3 An Account of Reward's Expressivity: The TASKQ and EXPRESSIONQ

Consider an onlooker, Alice, and an earnest learning agent, Bob, engaged in the interaction pictured in Figure 1. Suppose that Alice has a particular task in mind that she would like Bob to learn to solve, and that Alice constructs a reward function to incentivize Bob to pursue this task. Here, Alice is playing the role of "all of what we mean by goals and purposes" for Bob to pursue, with Bob playing the role of the standard reward-maximizing RL agent.

**Two Questions About Task.**    To give us leverage to study the expressivity of reward, it is useful to draw a distinction between two stages of this process: 1) Alice thinks of a task that she would like Bob to learn to solve, and 2) Alice creates a reward function (and perhaps chooses $\gamma$) that conveys the chosen task to Bob. We inspect these two separately, framed by the following two questions. The first we call the *task-definition question* (TASKQ) which asks: What *is* a task? The second we call the *task-expression question* (EXPRESSIONQ) which asks: Which learning signal can be used as a mechanism for expressing any task to Bob?

**Reward Answers The EXPRESSIONQ.**    We suggest that it may be useful to treat reward as an answer to the EXPRESSIONQ rather than the TASKQ. On this view, reward is treated as an expressive language for incentivizing reward-maximizing agents: Alice may attempt to translate any task into a reward function that incentivizes Bob to pursue the task, no matter which environment Bob inhabits, which task Alice has chosen, or how she has represented the task to herself. Indeed, it might be the case that Alice's knowledge of the task far exceeds Bob's representational or perceptual capacity. Alice may know every detail of the environment and define the task based on this holistic vantage, while Bob must learn to solve the task through interaction alone, relying only on a restricted class of functions for modeling and decision making.

Under this view, we can assess the expressivity of reward as an answer to the EXPRESSIONQ conditioned on how we answer the TASKQ. For example, if the TASKQ is answered in terms of natural language descriptions of desired states of affairs, then reward may fail to convey the chosen task due to the apparent mismatch in abstraction between natural language and reward (though some work has studied such a proposal [31, 62]).

## 3.1    Answers to the TASKQ: What is a Task?

In RL, tasks are often associated with a choice of goal, reward function ($R$), reward-discount pair ($R, \gamma$), or perhaps a choice of optimal policy (alongside those task types surveyed previously, such as LTL). However, it is unclear whether these constructs capture the entirety of what we mean by "task".

For example, consider the Russell and Norvig [42] grid world: A $4 \times 3$ grid with one wall, one terminal fire state, and one terminal goal state (pictured with a particular reward function in Figure 4a). In such an environment, how might we think about tasks? A standard view is that the task is to reach the goal as quickly as possible. This account, however, fails to distinguish between the *non*-optimal behaviors, such as the costly behavior of the agent moving directly into the fire and the neutral behavior of the agent spending its existence in the start state. Indeed, characterizing a task in terms of choice of $\pi^*$ or goal fails to capture these distinctions. Our view is that a suitably rich account of task should allow for the characterization of this sort of preference, offering the flexibility to scale from specifying only the desirable behavior (or outcomes) to an arbitrary ordering over behaviors (or outcomes).

In light of these considerations, we propose three answers to the TASKQ that can convey general preferences over behavior or outcome: 1) A set of acceptable policies, 2) A partial ordering over policies, or 3) A partial ordering over trajectories. We adopt these three as they can capture many kinds of task while also allowing a great deal of flexibility in the level of detail of the specification.

| Name | Notation | Generalizes | Constraints Induced by $\mathcal{T}$ |
|------|----------|-------------|--------------------------------------|
| SOAP | $\Pi_G$ | task-as-$\pi^*$ | equal: $V^{\pi_g}(s_0) = V^{\pi_{g'}}(s_0) > V^{\pi_b}(s_0), \forall_{\pi_g, \pi_{g'} \in \Pi_G, \pi_b \in \Pi_B}$ 
 range: $V^{\pi_g}(s_0) > V^{\pi_b}(s_0), \forall_{\pi_g \in \Pi_G, \pi_b \in \Pi_B}$ |
| PO | $L_\Pi$ | SOAP | $(\pi_1 \oplus \pi_2) \in L_\Pi \implies V^{\pi_1}(s_0) \oplus V^{\pi_2}(s_0)$ |
| TO | $L_{\tau,N}$ | task-as-goal | $(\tau_1 \oplus \tau_2) \in L_{\tau,N} \implies G(\tau_1; s_0) \oplus G(\tau_2; s_0)$ |

Table 1: A summary of the three proposed task types. We further list the constraints that determine whether a reward function *realizes* each task type in an MDP, where we take $\oplus$ to be one of '$<$', '$>$', or '$=$', and $G$ is the discounted return of the trajectory.

## 3.2 SOAPs, POs, and TOs

**(SOAP) Set Of Acceptable Policies.**    A classical view of the equivalence of two reward functions is based on the optimal policies they induce. For instance, Ng et al. [36] develop potential-based reward shaping by inspecting which shaped reward signals will ensure that the optimal policy is unchanged. Extrapolating, it is natural to say that for any environment $E$, two reward functions are equivalent if the optimal policies they induce in $E$ are the same. In this way, a task is viewed as a choice of optimal policy. As discussed in the grid world example above, this notion of task fails to allow for the specification of the quality of other behaviors. For this reason, we generalize task-as-optimal-policy to a *set of acceptable policies*, defined as follows.

**Definition 3.1.** *A set of acceptable policies (SOAP) is a non-empty subset of the deterministic policies,* $\Pi_G \subseteq \Pi$*, with $\Pi$ the set of all deterministic mappings from $\mathcal{S}$ to $\mathcal{A}$ for a given $E$.*

With one task type defined, it is important to address what it means for a reward function to properly *realize*, *express*, or *capture* a task in a given environment. We offer the following account.

**Definition 3.2.** *A reward function is said to realize a task $\mathcal{T}$ in an environment $E$ just when the start-state value (or return) induced by the reward function exactly adheres to the constraints of $\mathcal{T}$.*

Precise conditions for the realization of each task type are provided alongside each task definition, with a summary presented in column four of Table 1.

For SOAPs, we take the start-state value $V^\pi(s_0)$ to be the mechanism by which a reward function realizes a SOAP. That is, for a given $E$ and $\Pi_G$, a reward function $R$ is said to *realize* the $\Pi_G$ in $E$ when the start-state value function is optimal for all good policies, and strictly higher than the start-state value of all other policies. It is clear that SOAP strictly generalizes a task in terms of a choice of optimal policy, as captured by the SOAP $\Pi_G = \{\pi^*\}$.

We note that there are two natural ways for a reward function to realize a SOAP: First, each $\pi_g \in \Pi_G$ has *optimal* start-state value and all other policies are sub-optimal. We call this type *equal*-SOAP, or just SOAP for brevity. Alternatively, we might only require that the acceptable policies are each *near*-optimal, but are allowed to differ in start-state value so long as they are all better than *every* bad policy $\pi_b \in \Pi_B$. That is, in this second kind, there exists an $\epsilon \geq 0$ such that every $\pi_g \in \Pi_G$ is $\epsilon$-optimal in start-state value, $V^*(s_0) - V^{\pi_g}(s_0) \leq \epsilon$, while all other policies are worse. We call this second realization condition *range*-SOAP. We note that the range realization generalizes the equal one: Every equal-SOAP is a range-SOAP (by letting $\epsilon = 0$). However, there exist range-SOAPs that are expressible by Markov rewards that are *not* realizable as an equal-SOAP. We illustrate this fact with the following proposition. All proofs are presented in Appendix B.

**Proposition 3.1.** *There exists a CMP, $E$, and choice of $\Pi_G$ such that $\Pi_G$ can be realized under the range-SOAP criterion, but cannot be realized under the equal-SOAP criterion.*

One such CMP is pictured Figure 2b. Consider the SOAP $\Pi_G = \{\pi_{11}, \pi_{12}, \pi_{21}\}$: Under the equal-SOAP criterion, if each of these three policies are made optimal, any reward function will *also* make $\pi_{22}$ (the only bad policy) optimal as well. In contrast, for the range criterion, we can choose a reward function that assigns lower rewards to $a_2$ than $a_1$ in both states. In general, we take the equal-SOAP realization as canonical, as it is naturally subsumed by our next task type.

**(PO) Partial Ordering on Policies.** Next, we suppose that Alice chooses a *partial ordering* on the deterministic policy space. That is, Alice might identify a some great policies, some good, and some bad policies to strictly avoid, and remain indifferent to the rest. POs strictly generalize equal SOAPs, as any such SOAP is a special choice of PO with only two equivalence classes. We offer the following definition of a PO.

**Definition 3.3.** *A policy order (PO) of the deterministic policies $\Pi$ is a partial order, denoted $L_\Pi$.*

As with SOAPs, we take the start-state value $V^\pi(s_0)$ induced by a reward function $R$ as the mechanism by which policies are ordered. That is, given $E$ and $L_\Pi$, we say that a reward function $R$ *realizes* $L_\Pi$ in $E$ if and only if the resulting MDP, $M = (E, R)$, produces a start-state value function that orders $\Pi$ according to $L_\Pi$.

**(TO) Partial Ordering on Trajectories.** A natural generalization of goal specification enriches a notion of task to include the details of how a goal is satisfied—that is, for Alice to relay some preference over trajectory space [63], as is done in preference based RL [64]. Concretely, we suppose Alice specifies a partial ordering on length $N$ trajectories of $(s, a)$ pairs, defined as follows.

**Definition 3.4.** *A trajectory ordering (TO) of length $N \in \mathbb{N}$ is a partial ordering $L_{\tau,N}$, with each trajectory $\tau$ consisting of $N$ state–action pairs, $\{(s_0, a_0), \dots, (a_{N-1}, s_{N-1})\}$, with $s_0$ the start state.*

As with PO, we say that a reward function realizes a trajectory ordering $L_{\tau,N}$ if the ordering determined by each trajectory's cumulative discounted $N$-step return from $s_0$, denoted $G(\tau; s_0)$, matches that of the given $L_{\tau,N}$. We note that trajectory orderings can generalize goal-based tasks at the expense of a larger specification. For instance, a TO can convey the task, "Safely reach the goal in less than thirty steps, or just get to the subgoal in less than twenty steps."

**Recap.** We propose to assess the expressivity of reward by first answering the TASKQ in terms of SOAPs, POs, or TOs, as summarized by Table 1. We say that a task $\mathcal{T}$ is *realized* in an environment $E$ under reward function $R$ if the start-state value function (or return) produced by $R$ imposes the constraints specified by $\mathcal{T}$, and are interested in whether reward can always realize a given task in any choice of $E$. We make a number of assumptions along the way, including: (1) Reward functions are Markov and deterministic, (2) Policies of interest are deterministic, (3) The environment is a finite CMP, (4) $\gamma$ is part of the environment, (5) We ignore reward's role in shaping the *learning process*, (6) Start-state value or return is the appropriate mechanism to determine if a reward function realizes a given task. Relaxation of these assumptions is a critical direction for future work.

## 4 Analysis: The Expressivity of Markov Reward

With our definitions and objectives in place, we now present our main results.

### 4.1 Express SOAPs, POs, and TOs

We first ask whether reward can always realize a given SOAP, PO, or TO, for an arbitrary $E$. Our first result states that the answer is "no"—there are tasks that cannot be realized by any reward function.

**Theorem 4.1.** *For each of SOAP, PO, and TO, there exist $(E, \mathcal{T})$ pairs for which no Markow reward function realizes $\mathcal{T}$ in $E$.*

Thus, reward is incapable of capturing certain tasks. What tasks are they, precisely? Intuitively, inexpressible tasks involve policies or trajectories that *must* be correlated in *value* in an MDP. That is, if two policies are nearly identical in behavior, it is unlikely that reward can capture the PO that places them at opposite ends of the ordering. A simple example is the "always move the same direction" task in a grid world, with state defined as an $(x, y)$ pair. The SOAP $\Pi_G = \{\pi_\leftarrow, \pi_\uparrow, \pi_\rightarrow, \pi_\downarrow\}$ conveys this task, but no Markov reward function can make these policies strictly higher in value than all others.

**Example: Inexpressible SOAPs.** Observe the two CMPs pictured in Figure 2, depicting two kinds of inexpressible SOAPs. On the left, we consider the SOAP $\Pi_G = \{\pi_{21}\}$, containing only the policy that executes $a_2$ in the left state ($s_0$), and $a_1$ in the right ($s_1$). This SOAP is inexpressible through reward, but only because reward cannot distinguish the start-state value of $\pi_{21}$ and $\pi_{22}$ since the policies differ only in an unreachable state. This is reminiscent of Axiom 5 from Pitis [40], which

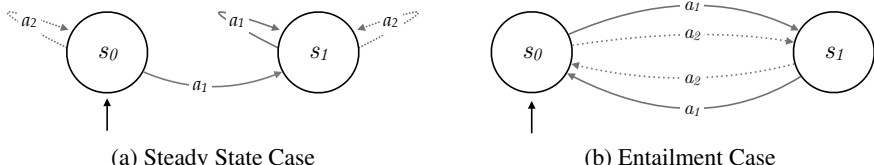

(a) Steady State Case      (b) Entailment Case

Figure 2: Two CMPs in which there is a SOAP that is not expressible under any Markov reward function. On the left, $\Pi_G = \{\pi_{21}\}$ is not realizable, as $\pi_{21}$ can not be made better than $\pi_{22}$ because $s_1$ is never reached. On the right, the XOR-like-SOAP, $\Pi_G = \{\pi_{12}, \pi_{21}\}$ is not realizable: To make these two policies optimal, it is entailed that $\pi_{22}$ and $\pi_{11}$ must be optimal, too.

explicitly excludes preferences of this sort. On the right, we find a more interesting case: The chosen SOAP is similar to the XOR function, $\Pi_G = \{\pi_{12}, \pi_{21}\}$. Here, the task requires that the agent choose each action in exactly one state. However, there cannot exist a reward function that makes *only* these policies optimal, as by consequence, both policies $\pi_{11}$ and $\pi_{22}$ *must* be optimal as well.

Next, we show that Theorem 4.1 is not limited to a particular choice of transition function or $\gamma$.

**Proposition 4.2.** *There exist choices of $E_{\neg T} = (\mathcal{S}, \mathcal{A}, \gamma, s_0)$ or $E_{\neg \gamma} = (\mathcal{S}, \mathcal{A}, T, s_0)$, together with a task $\mathcal{T}$, such that there is no $(T, R)$ pair that realizes $\mathcal{T}$ in $E_{\neg T}$ or $(R, \gamma)$ in $E_{\neg \gamma}$.*

This result suggests that the scope of Theorem 4.1 is actually quite broad—even if the transition function or $\gamma$ are taken as part of the reward specification, there are tasks that cannot be expressed. We suspect there are ways to give a precise characterization of *all* inexpressible tasks from an axiomatic perspective, which we hope to study in future work.

## 4.2 Constructive Algorithms: Task to Reward

We now analyze how to determine whether an appropriate reward function can be constructed for any $(E, \mathcal{T})$ pair. We pose a general form of the reward-design problem [34, 51, 9] as follows.

**Definition 4.1.** *The REWARDDESIGN problem is: **Given** $E = (\mathcal{S}, \mathcal{A}, T, \gamma, s_0)$, and a $\mathcal{T}$, **output** a reward function $R_{alice}$ that ensures $\mathcal{T}$ is realized in $M = (E, R_{alice})$.*

Indeed, for all three task types, there is an efficient algorithm for solving the reward-design problem.

**Theorem 4.3.** *The REWARDDESIGN problem can be solved in polynomial time, for any finite $E$, and any $\mathcal{T}$, so long as reward functions with infinitely many outputs are considered.*

Therefore, for *any* choice of finite CMP, $E$, and a SOAP, PO, or TO, we can find a reward function that perfectly realizes the task in the given environment, if such a reward function exists. Each of the three algorithms are based on forming a linear program that matches the constraints of the given task type, which is why reward functions with infinitely many outputs are required. Pseudo-code for SOAP-based reward design is presented in Algorithm 1. Intuitively, the algorithms compute the discounted expected-state visitation distribution for a collection of policies; in the case of SOAP, for instance, these policies include $\Pi_G$ and what we call the "fringe", the set of policies that differ from a $\pi_g \in \Pi_G$ by exactly one action. Then, we use these distributions to describe linear inequality constraints ensuring that the start-state value of the good policies are better than those of the fringe.

As highlighted by Theorem 4.1 there are SOAPs, POs, and TOs that are *not realizable*. Thus, it is important to determine how the algorithms mentioned in Theorem 4.3 will handle such cases. Our next corollary illustrates that the desirable outcome is achieved: For any $E$ and $\mathcal{T}$, the algorithms will output a reward function that realizes $\mathcal{T}$ in $E$, or output '$\perp$' when no such function exists.

**Corollary 4.4.** *For any task $\mathcal{T}$ and environment $E$, deciding whether $\mathcal{T}$ is expressible in $E$ is solvable in polynomial time.*

Together, Theorem 4.1 and Theorem 4.3 constitute our main results: There are environment–task pairs in which Markov reward cannot express the chosen task for each of SOAPs, POs, and TOs. However, there are efficient algorithms for deciding whether a task is expressible, and for constructing the realizing reward function when it exists. We will study the use of one of these algorithms in Section 5, but first attend to other aspects of reward's expressivity.

**Algorithm 1** SOAP Reward Design

INPUT: $E = (\mathcal{S}, \mathcal{A}, T, \gamma, s_0), \Pi_G$.
OUTPUT: $R$, or $\perp$.

```
 1: Π_fringe = compute_fringe(Π_G)
 2: for π_{g,i} ∈ Π_G do                                  ▷ Compute state-visitation distributions.
 3:     ρ_{g,i} = compute_exp_visit(π_{g,i}, E)

 4: for π_{f,i} ∈ Π_fringe do
 5:     ρ_{f,i} = compute_exp_visit(π_{f,i}, E)

 6: C_eq = {}                                             ▷ Make Equality Constraints.
 7: for π_{g,i} ∈ Π_G do
 8:     C_eq.add(ρ_{g,0}(s_0) · X = ρ_{g,i}(s_0) · X)

 9: C_ineq = {}                                           ▷ Make Inequality Constraints.
10: for π_{f,j} ∈ Π_fringe do
11:     C_ineq.add(ρ_{f,j}(s_0) · X + ε ≤ ρ_{g,0}(s_0) · X)

12: R_out, ε_out = linear_programming(obj. = max ε, constraints = C_ineq, C_eq)      ▷ Solve LP.

13: if ε_out > 0 then                                     ▷ Check if successful.
        return R_out
14: else
        return ⊥
```

### 4.3   Other Aspects of Reward's Expressivity

We next briefly summarize other considerations about the expressivity of reward. As noted, Theorem 4.3 requires the use of a reward function that can produce infinitely many outputs. Our next result proves this requirement is strict for efficient reward design.

**Theorem 4.5.** *A variant of the* REWARDDESIGN *problem with finite reward outputs is NP-hard.*

We provide further details about the precise problem studied in Appendix B. Beyond reward functions with finitely-many outputs, we are also interested in extensions of our results to multiple *environments*. We next present a positive result indicating our algorithms can extend to the case where Alice would like to design a reward function for a single task across multiple environments.

**Proposition 4.6.** *For any SOAP, PO, or TO, given a finite set of CMPs, $\mathcal{E} = \{E_1, \ldots, E_n\}$, with shared state–action space, there exists a polynomial time algorithm that outputs one reward function that realizes the task (when possible) in all CMPs in $\mathcal{E}$.*

A natural follow up question to the above result asks whether task realization is *closed* under a set of CMPs. Our next result answers this question in the negative.

**Theorem 4.7.** *Task realization is not closed under sets of CMPs with shared state-action space. That is, there exist choices of $\mathcal{T}$ and $\mathcal{E} = \{E_1, \ldots, E_n\}$ such that $\mathcal{T}$ is realizable in each $E_i \in \mathcal{E}$ independently, but there is not a single reward function that realizes $\mathcal{T}$ in all $E_i \in \mathcal{E}$ simultaneously.*

Intuitively, this shows that Alice must know precisely which environment Bob will inhabit if she is to design an appropriate reward function. Otherwise, her uncertainty over $E$ may prevent her from designing a realizing reward function. We foresee iterative extensions of our algorithms in which Alice and Bob can react to one another, drawing inspiration from repeated IRL by Amin et al. [4].

## 5   Experiments

We next conduct experiments to shed further light on the findings of our analysis. Our focus is on SOAPs, though we anticipate the insights extend to POs and TOs as well with little complication. In the first experiment, we study the fraction of SOAPs that are expressible in small CMPs as we vary aspects of the environment or task (Figure 3). In the second, we use one algorithm from Theorem 4.3

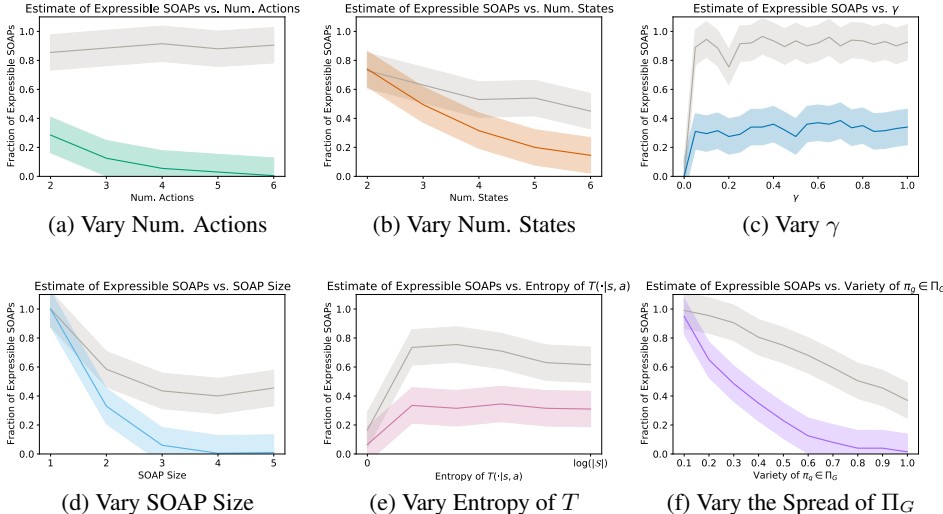

Figure 3: The approximate fraction of SOAPs that are expressible by reward in CMPs with a handful of states and actions, with 95% confidence intervals. In each plot, we vary a different parameter of the environment or task to illustrate how this change impacts the expressivity of reward, showing both equal (color) and range (grey) realization of SOAP.

to design a reward function, and contrast learning curves under a SOAP-designed reward function compared to standard rewards. Full details about the experiments are found in Appendix C.

**SOAP Expressivity.** First, we estimate the fraction of SOAPs that are expressible in small environments. For each data point, we sample 200 random SOAPs and run Algorithm 1 described by Theorem 4.3 to determine whether each SOAP is realizable in the given CMP. We ask this question for both the equal (color) variant of SOAP realization and the range (grey) variant. We inspect SOAP expressivity as we vary six different characteristics of $E$ or $\Pi_G$: The number of actions, the number of states, the discount $\gamma$, the number of good policies in each SOAP, the Shannon entropy of $T$ at each $(s, a)$ pair, and the "spread" of each SOAP. The spread approximates average edit distance among policies in $\Pi_G$ determined by randomly permuting actions of a reference policy by a coin weighted according to the value on the x-axis. We use the same set of CMPs for each environment up to any deviations explicitly made by the varied parameter (such as $\gamma$ or entropy). Unless otherwise stated, each CMP has four states and three actions, with a fixed but randomly chosen transition function.

Results are presented in Figure 3. We find that our theory is borne out in a number of ways. First, as Theorem 4.1 suggests, we find SOAP expressivity is strictly less than one in nearly all cases. This is evidence that inexpressible tasks are not only found in manufactured corner cases, but rather that expressivity is a spectrum. We further observe—as predicted by Proposition 3.1—clear separation between the expressivity of range-SOAP (grey) vs. equal-SOAP (color); there are many cases where we can find a reward function that makes the good policies *near* optimal and better than the bad, but cannot make those good policies all *exactly* optimal. Additionally, several trends emerge as we vary the parameter of environment or task, though we note that such trends are likely specific to the choice of CMP and may not hold in general. Perhaps the most striking trend is in Figure 3f, which shows a decrease in expressivity as the SOAPs become more *spread out*. This is quite sensible: A more spread out SOAP is likely to lead to more entailments of the kind discussed in Figure 2b.

**Learning with SOAP-designed Rewards.** Next, we contrast the learning performance of Q-learning under a SOAP-designed reward function (visualized in Figure 4a) with that of the regular goal-based reward in the Russell and Norvig [42] grid world. In this domain, there is 0.35 slip probability such that, on a 'slip' event, the agent randomly applies one of the two orthogonal action effects. The regular goal-based reward function provides $+1$ when the agent enters the terminal flag cell, and $-1$ when the agent enters the terminal fire cell. The bottom left state is the start-state, and the black cell is an impassable wall.

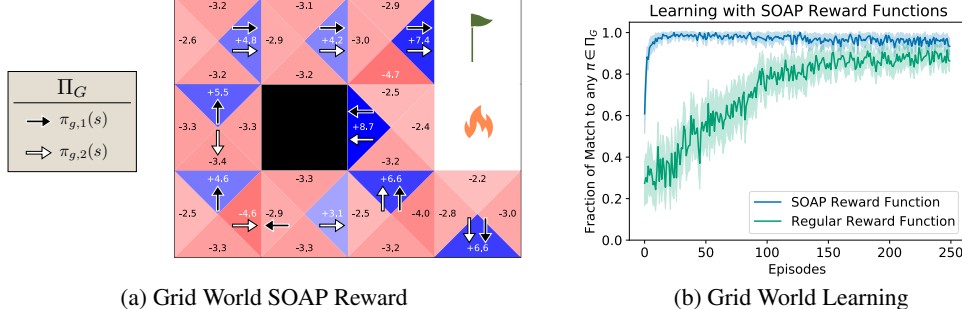

|                              |                              |
| :--------------------------: | :--------------------------: |
| (a) Grid World SOAP Reward   | (b) Grid World Learning      |

Figure 4: A SOAP-designed reward function (left) and the resulting learning curves (right) for Q-learning compared to the traditional reward function for the Russell and Norvig [42] grid world. Each series presents average performance over 50 runs of the experiment with 95% confidence intervals.

Results are presented in Figure 4. On the right, we present a particular kind of learning curve contrasting the performance of Q-learning with the SOAP reward (blue) and regular reward (green). The y-axis measures, at the end of each episode, the average (inverse) minimum edit distance between Q-learning's greedy policy and any policy in the SOAP. Thus, when the series reaches 1.0, Q-learning's greedy policy is identical to one of the two SOAP policies. We first find that Q-learning is able to quickly learn a $\pi_g \in \Pi_G$ under the designed reward function. We further observe that the typical reward does not induce a perfect match in policy—at convergence, the green curve hovers slightly below the blue, indicating that the default reward function is incentivizing different policies to be optimal. This is entirely sensible, as the two SOAP policies are extremely cautious around the fire; they choose the orthogonal (and thus, safe) action in fire-adjacent states, relying on slip probability to progress. Lastly, as expected given the amount of knowledge contained in the SOAP, the SOAP reward function allows Q-learning to rapidly identify a good policy compared to the typical reward.

## 6   Conclusion

We have here investigated the expressivity of Markov reward, framed around three new accounts of task. Our main results show that there exist choices of task and environment in which Markov reward cannot express the chosen task, but there are efficient algorithms that decide whether a task is expressible *and* construct a reward function that captures the task when such a function exists. We conclude with an empirical examination of our analysis, corroborating the findings of our theory. We take these to be first steps toward understanding the full scope of the reward hypothesis.

There are many routes forward. A key direction moves beyond the task types we study here, and relaxes our core assumptions—the environment might not be a finite CMP, Alice may not know the environment precisely, reward may be a function of history, or Alice may not know how Bob represents state. Along similar lines, a critical direction incorporates how reward impacts Bob's *learning dynamics* rather than start-state value. Further, we note the potential relevance to the recent *reward-is-enough* hypothesis proposed by Silver et al. [44]; we foresee pathways to extend our analysis to examine this newer hypothesis, too. For instance, in future work, it is important to assess whether reward is capable of inducing the right kinds of attributes of cognition, not just behavior.

## Acknowledgments and Disclosure of Funding

The authors would like to thank André Barreto, Diana Borsa, Michael Bowling, Wilka Carvalho, Brian Christian, Jess Hamrick, Steven Hansen, Zac Kenton, Ramana Kumar, Katrina McKinney, Rémi Munos, Matt Overlan, Hado van Hasselt, and Ben Van Roy for helpful discussions. We would also like to thank the anonymous reviewers for their thoughtful feedback, and Brendan O'Donoghue for catching a typo in the appendix. Michael Littman was supported in part by funding from DARPA L2M, ONR MURI, NSF FMitF, and NSF RI.

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
