# Appendix: On the Expressivity of Markov Reward

**David Abel**
DeepMind
dmabel@deepmind.com

**Will Dabney**
DeepMind
wdabney@deepmind.com

**Anna Harutyunyan**
DeepMind
harutyunyan@deepmind.com

**Mark K. Ho**
Department of Computer Science
Princeton University
mho@princeton.edu

**Michael L. Littman**
Department of Computer Science
Brown University
mlittman@cs.brown.edu

**Doina Precup**
DeepMind
doinap@deepmind.com

**Satinder Singh**
DeepMind
baveja@deepmind.com

## A  Anticipated Questions

We first address questions that might arise in response to the main text.

*(Q1) What does it mean for Bob to \*solve\* one of these tasks? That is, if Alice chooses a SOAP, PO, or TO for Bob to learn to solve, when can Alice determine Bob has solved the task?*

A: Bob can be said to be doing better on a given task if his behavior improves, as is typical in evaluating behavior under reward. The difference with SOAPs, POs, and TOs is that we measure improvement relative to the task rather than reward. For instance, given a SOAP, we might say that Bob has solved the task once he has found one of the good policies, and we might measure Bob's progress on a task in terms of the distance of his greedy policy to one of the good policies (as done in our learning experiments). The same reasoning applies to POs and TOs: Bob is doing better on a task in so far as his greedy policy (or trajectories) is (are) higher up the ordering.

*(Q2) These notions of inexpressibility all come about due to the Markov restriction on reward functions. That is, the studied reward functions must be a function of $s$, $(s, a)$, or $(s, a, s')$. But, what about history-based reward functions?*

A: Indeed, as discussed in our introduction, our goal is to examine the expressivity of Markov rewards in the context of finite MDPs. We assume the environment is fixed and given to Alice with the state and action spaces already determined. While it is sensible to consider history-based rewards, this opens up new considerations: Must the state space also change so as to retain the Markov property? Instead, we suggest that for a given CMP, it is natural to be interested in Markov rewards, but acknowledge the importance of going beyond such functions. As discussed in the main text, we suspect that there is a coherent account of which tasks are and are not expressible as a consequence of some of the axioms for rationality. We hope to study these directions in future work.

*(Q3) Why restrict attention to SOAPs, POs, and TOs?*

A: First, we recognize these do not necessarily capture all of what we hope to convey to learning agents. It is an important next step in our work to enrich the analysis with more general objectives. Still, we believe that these each represent an interesting, relatively general, and concrete template for what a task might look like. They are quite flexible: SOAPs can be simple while POs and TOs can be complex.

35th Conference on Neural Information Processing Systems (NeurIPS 2021).

*(Q4) Why restrict to the start-state value?*

A: We adopt start-state value due to its simplicity. Other considerations might be: (1) The *expected* value under some chosen distribution, or (2) That the constraint hold over all states (so, for SOAPs, each $\pi_g$ is better than each $\pi_b$ in value for all states). We note that the former case is identical to start-state value, as we can always add a start-state to any CMP where all actions lead to the desired next-state distribution in $T$. The latter case is slightly more complicated, so we chose not to focus on it as we prefer the simplicity of the start-state case. However, we note that many inexpressible tasks under the start-state criterion remain inexpressible under the "all-state" criterion (such as the XOR example from Figure 2b).

..........................

## B Proofs

We next restate each central result, and present its proof.

**Proposition 3.1.** *There exists a CMP, E, and choice of $\Pi_G$ such that $\Pi_G$ can be realized under the range-SOAP criterion, but cannot be realized under the equal-SOAP criterion.*

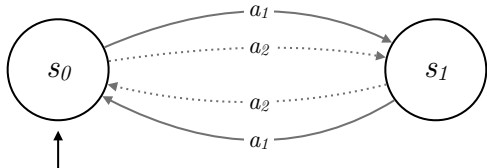

Figure B.1: A CMP that separates the two kinds of SOAP realizations.

*Proof of Proposition 3.1.*

Consider the example in Figure B.1 and the SOAP $\Pi_G = \{\pi_{11}, \pi_{21}, \pi_{12}\}$. This $\Pi_G$ indicates that all policies are acceptable except the policy that always takes $a_2$. That is, the policy subscripts denote which actions each policy takes in each state ($\pi_{12}$ means $a_1$ in $s_0$, $a_2$ in $s_1$).

First, let us note that range-SOAP is realizable: The listed rewards allow for each policy in $\Pi_G$ to obtain $\frac{\text{RMAX}/2}{1-\gamma}$ value or better, while $\pi_{22}$ achieves zero. Letting $\varepsilon = \text{VMAX}/2$, this choice of rewards satisfies the criteria, and the $\Pi_G$ is $\varepsilon$-realized in the given MDP.

Next, note that there can exist no other choice of rewards that realize the equal-SOAP. That is, such that $V^{\pi_{11}}(s_0) = V^{\pi_{12}}(s_0) = V^{\pi_{21}}(s_0) > V^{\pi_{22}}(s_0)$. This fact is a consequence of the tie in values between the policies. Here, we see that any choice of rewards that makes $V^{\pi_{12}}(s_0) = V^{\pi_{21}}(s_0)$ will also give $\pi_{22}$ that same value. Thus, the given $\Pi_G$ is unrealizable under equal-SOAP. $\square$

**Theorem 4.1.** *For each of SOAP, PO, and TO, there exist $(E, \mathscr{T})$ pairs for which no reward function realizes $\mathscr{T}$ in E.*

*Proof of Theorem 4.1.*

We proceed by proving the existence of a pair $(E, \mathscr{T})$, for each of $\mathscr{T}$ as a SOAP, PO, or TO. Indeed, we find that the simple XOR case is inexpressible for all three task types.

**SOAP.** For SOAP, we consider the CMP with two states and two actions from Figure B.1, and the SOAP $\Pi_G = \{\pi_{12}, \pi_{21}\}$. That is, the chosen task is for the learning agent to find a policy that chooses each action in exactly one state. Here, we find that any Markov reward function that makes $a_1$ optimal in the left state will, by consequence, make $a_1$ an optimal action no matter what is done in other states. In other words, we cannot assign rewards to

$(s, a)$ pairs so that an action's optimality depends on *which* optimal action is taken in the other state. Thus, all choices of Markov reward function that make $\{\pi_{12}, \pi_{21}\}$ optimal will also make $\{\pi_{11}, \pi_{22}\}$ optimal, too.

**PO.** Since the given SOAP is a special case of a PO, we have already identified a given inexpressible PO.

**TO.** For TO, for simplicity we consider the same CMP. We let $N = 2$, and suppose that the desired trajectory ordering is over state-action pairs, giving rise to a set of good trajectories, and a set of bad trajectories:

$$L_{\tau,N} := \{\tau_G, \tau_B\}, \tag{B.1}$$
$$\tau_G = \{\{(s_0, a_1), (s_1, a_2)\}, \{(s_0, a_2), (s_1, a_1)\}\}, \tag{B.2}$$
$$\tau_B = \{\{(s_0, a_1), (s_1, a_1)\}, \{(s_0, a_2), (s_1, a_2)\}\}. \tag{B.3}$$

The same reasoning from the above cases applies: We cannot make the good trajectories strictly higher in return than the bad trajectories. $\qquad\square$

**Proposition 4.2.** *There exist choices of* $E_{\neg T} = (\mathcal{S}, \mathcal{A}, \gamma, s_0)$ *or* $E_{\neg\gamma} = (\mathcal{S}, \mathcal{A}, T, s_0)$*, together with a task* $\mathcal{T}$*, such that there is no* $(T, R)$ *pair that realizes* $\mathcal{T}$ *in* $E_{\neg T}$ *or* $(R, \gamma)$ *in* $E_{\neg\gamma}$*.*

***Proof of Proposition 4.2.***

The running XOR example is actually inexpressible for *all* choices of $T$, or of $\gamma$. That is, there is no way to make $\pi_{12}$ and $\pi_{21}$ strictly better than both $\pi_{22}$ *and* $\pi_{11}$ by varying $\gamma$ or $T$. Such examples likely exist for any choice of $\mathcal{S}$ and $\mathcal{A}$ of size greater than one. $\qquad\square$

**Theorem 4.3.** *The* REWARDDESIGN *problem can be solved in polynomial time, for any finite* $E$*, and any SOAP, PO, or TO, so long as a reward-function family with infinitely many outputs is used.*

***Proof of Theorem 4.3.***

We proceed by providing constructive algorithms for each of SOAP, PO, or TO. All three are based on similar applications of a linear program (LP), though there is nuance that separates them. We present each as a Lemma (Lemma B.1, Lemma B.2, Lemma B.3), which together constitute the proof of this Theorem. $\qquad\square$

**Lemma B.1.** *The SOAP variant of* REWARDDESIGN *can be solved in polynomial time.*

***Proof of Lemma B.1.***

We proceed by constructing a linear program (LP) whose solution is the desired reward function (or correctly outputs that there is no such reward function). Specifically, note that we want to choose a reward function so that all the policies in $\Pi_G$ have strictly higher start-state value than all the policies not in the set. We present the proof through five observations.

**First,** observe that any reward function that will induce the desired ordering ensures that the optimal policy $\pi^*$ is in the set $\Pi_G$. This is true since $\pi^*$ (under the chosen reward function) is better than all policies. So it is better than all policies not in $\Pi_G$.

**Second,** note that the set $\Pi_G$ is well connected in the following sense. Let a step in policy space from some reference policy $\pi_{\text{ref}}$ to be a move to any other deterministic policy that differs from $\pi_{\text{ref}}$ in exactly one state. Then, for any pair of policies in $\Pi_G$, there must be a sequence of policies in $\Pi_G$, each one step apart from the next, from one to the other. This follows from the policy-improvement theorem: we can get from any policy to an optimal

policy in a sequence of policies such that each policy (1) is one step from the previous one and (2) strictly dominates the previous one. Since any policy that strictly dominates a policy in $\Pi_G$ must be better than the policy in $\Pi_G$, it must also be in $\Pi_G$ (if the problem constraint is satisfied). That means if we choose two policies in $\Pi_G$, $\pi_1$ and $\pi_2$, both can reach $\pi^*$ in a sequence of single steps while staying within $\Pi_G$. Since steps are symmetric, $\Pi_G$ is connected. The connected set of policies in $\Pi_G$ has a "fringe" $\Pi_{\text{fringe}}$—a set of policies not in $\Pi_G$ that are one step from a policy in $\Pi_G$.

**Third,** for the constraints of the problem to be satisfied, every policy $\pi_g \in \Pi_G$ must be strictly better than every policy $\pi_f \in \Pi_{\text{fringe}}$.

**Fourth,** observe that $|\Pi_{\text{fringe}}| <= |\mathcal{A}||\Pi_G|$, so $\Pi_{\text{fringe}}$ is polynomial sized.

**Fifth,** we can construct a polynomial-sized LP that expresses that every policy $\pi_g \in \Pi_G$ is strictly better than every policy $\pi_f \in \Pi_{\text{fringe}}$. Note that the direct way to build this LP has a "strictly better than" comparison between each policy $\pi_f \in \Pi_{\text{fringe}}$ and each policy $\pi \in \Pi_G$. That's at most $|\mathcal{A}||\Pi_G|^2$ inequalities.

We now tie the above observations together to show that the solution to this LP solves the constraints of the problem. That is, the reward function returned makes it so every policy $\pi_g \in \Pi_G$ is strictly better than every policy not in $\Pi_G$, *and* no valid reward function is excluded (so, if a solution exists, it will be found).

The argument that no valid reward function is excluded is simply because the set of constraints in the LP is a subset of the defining constraints of the problem. Specifically, the LP constrains the policies inside $\Pi_G$ to be strictly better than the ones on the fringe instead of all policies not in $\Pi_G$.

The argument that only constraining the values on the fringe automatically constrains all the others proceeds as follows. First, with respect to the returned reward function, there is some optimal policy $\pi^*$. That policy $\pi^*$ must be in $\Pi_G$. To see why, let us assume it is not. That means there is a sequence of improving steps that turn a policy in $\Pi_G$ to $\pi^*$ (currently assumed to be out of $\Pi_G$). But, any such sequence must go through the fringe, and we constrained the fringe so that all of the policies in $\Pi_G$ are strictly better than them. So, $\pi^*$ must be in $\Pi_G$. Next, we know that all policies in $\Pi_G$ must be strictly better than the policies not in $\Pi_G$. To see why, consider an "improving" path from some policy $\pi_b \notin \Pi_G$ to $\pi^*$. Since $\pi^*$ is in $\Pi_G$, we know this path must go through some policy $\pi_f \in \Pi_{\text{fringe}}$. Since it's an improving path, that means $\pi_f$ is better than $\pi_b$. But, every policy in $\Pi_G$ is strictly better than $\pi_f$, so it must also be strictly better than $\pi_b$. $\qquad\square$

**Lemma B.2.** *The PO variant of* REWARDDESIGN *can be solved in polynomial time.*

***Proof of Lemma B.2.***

We proceed by constructing a procedure that calls a linear program whose answer is the reward function that induces the given $L_\Pi$ in $M$, or the procedure correctly outputs that there is no such $R$.

Consider the set of policies in $\Pi$, numbered $\pi_1, \ldots, \pi_i, \ldots, \pi_N$. Note that the value of $\pi_i$ in $s_0$ can be computed in terms of the expected reward under the policy's discounted expected state-action visitation distribution. That is, for each $\pi_i$, let

$$\rho_i(s, a) := \sum_{t=0}^{\infty} \gamma^t \Pr(s_t = s, a_t = a \mid s_0, \pi_i). \tag{B.4}$$

Since the given MDP is assumed to have finite state-action space, note that $\rho_i$ may be interpreted as a vector whose elements correspond to $\rho_i(s_0, a_0), \rho_i(s_0, a_1)$, and so on.

The value of $\pi_i$ under a given reward function $R$ (which may also be interpreted as a vector) is then produced by the dot product $R \cdot \rho_i$.

Given $L_\Pi$, we want to find an $R$ that ensures a set of linear constraints hold:

$$R \cdot \rho_0 \geq R \cdot \rho_1 \geq \ldots . \tag{B.5}$$

Note that the trivial reward function, $R_\emptyset : s \mapsto 0$, is a solution to the above linear program. However, we can ensure some minimal increment improvement of $\epsilon$ for non-tying policies, where

$$R \cdot \rho_0 \geq R \cdot \rho_1 + \epsilon \geq \ldots . \tag{B.6}$$

This $\epsilon$ minimal increment is sufficient to separate policies with tying scores and avoids the degenerate solution of $R_\emptyset$, so long as there are infinitely many reward outputs feasible.

Note that the input is of size $N$, where $N$ is the number of constraints imposed on the policy ordering. In the worst case, $N = |L_\Pi| \leq |\mathcal{A}|^{|\mathcal{S}|}$. If there are fewer constraints than either $|\mathcal{A}|$ or $|\mathcal{S}|$, then $N = \max\{|\mathcal{S}|, |\mathcal{A}|\}$. The amount of computational work required is split across two steps:

1. $\tilde{O}(N)$: Compute $\rho_i$ for each policy $\pi_1 \ldots \pi_N$.

2. $O(N^3)$: Formulate and solve the above linear program.

Thus, since the described linear program can be constructed in polynomial time outputs a reward function that induces the given $L_\Pi$, we conclude the PO case. □

**Lemma B.3.** *The TO variant of* REWARDDESIGN *can be solved in polynomial time.*

***Proof of Lemma B.3.***

The algorithm follows the same construction as those catered toward SOAPs and POs, but is in fact much simpler.

We can form linear inequality constraints on the return of two trajectories as follows. First recall that the $N$-step discounted return of a trajectory $\tau$ is

$$G(\tau; s_0) = \sum_{i=0}^{N-1} \gamma^i R(s_i, a_i), \tag{B.7}$$

assuming reward is a function of state and action for simplicity. Note that because reward functions are assumed to be deterministic, the quantity $G(\tau; s_0)$ is not a random variable.

Now, given two trajectories,

$$\tau_i = \{(s_0^{(i)}, a_0^{(i)}), \ldots, (s_{N-1}^{(i)}, a_{N-1}^{(i)})\}, \quad \tau_j = \{(s_0^{(j)}, a_0^{(j)}), \ldots, (s_{N-1}^{(j)}, a_{N-1}^{(j)})\}, \tag{B.8}$$

note that we can express linear inequality constraints as follows,

$$\tau_i \cdot R - \tau_j \cdot R \leq \epsilon, \tag{B.9}$$

where $R$ is a length $N$ state-action vector, and $\epsilon \in \mathbb{Q}_{\geq 0}$ is a slack variable to be maximized as part of the optimization. Inequality constraints follow the same structure, only simpler. By the same reasoning that underlies the construction of the SOAP and PO based algorithms, the above set of constraints define a linear program whose solution is the realizing reward function, if it exists. □

**Corollary 4.4.** *For any task $\mathscr{T}$ and environment $E$, deciding whether $\mathscr{T}$ is expressible in $E$ is solvable in polynomial time*

***Proof of Corollary 4.4.***

For each of PO and TO, the constraints we construct define precisely the space of constraints that constitute the given task. Thus, since linear programming will find a solution for the given constraint set, we know that these two forms of the algorithm will also correctly decide when no reward function exists.

The SOAP case is slightly more involved, but still relatively straightforward. We note that the policy fringe, $\Pi_{\text{fringe}}$, is a subset of $\Pi_B$, since $\Pi_{\text{fringe}}$ consists of some policies *not* in $\Pi_G$ by construction. This means that the constraints produced that separate each good policy from a fringe policy in value are a subset of the true constraints (those that separate each $\pi_g \in \Pi_G$ from each $\pi_b \in \Pi_B$). Hence, since constraint relaxations of this kind have the property that they do not exclude solutions, we conclude that our proposed linear program will correctly determine when no satisfying reward function exists. □

Next, we provide further details on Theorem 4.5 that examines reward design when only finitely many reward outputs may be used. As noted in the main text, Theorem 4.3 requires that Alice is allowed to design a reward function that can produce infinitely many outputs. It is natural to wonder whether this requirement is strict. Theorem 4.5 answers this question in the affirmative, by proving that the following decision problem is hard.

**Definition B.1.** *The* FINITE-PO-REWARDDECISION *problem is defined as follows:* **Given** $E = (\mathcal{S}, \mathcal{A}, T, \gamma, s_0)$, *and a set of* $k$ *policy inequalities* ($\pi_{x_i} < \pi_{y_i}$), **output** *True iff there is a reward function* $R(s')$ *that induces the given policy inequalities.*

Note that this formulation focuses on POs, and on reward as a function of next-state. Unfortunately, we find this problem is NP-hard, showing that for reward design to be efficient, infinitely many reward outputs are needed.

**Theorem 4.5.** *The* FINITE-PO-REWARDDECISION *problem is NP-hard.*

***Proof of Theorem 4.5.***

We assume every $T(s' \mid s, a)$ is expressed as a rational number. We also assume that all policies are deterministic, Markov policies, although results should extend to stochastic policies with rational probabilities as well.

The binary PO problem is the same, but it insists that every $R(s') \in \{0, 1\}$ for all $s'$ in the returned reward function.

**Observation 1:** The binary PO decision problem is in NP. We can guess an assignment of $R(s, a)$ to either 0 or 1, then evaluate in inequality using linear equation solving as policy evaluation.

We show that the binary PO decision problem can be used to decide the NP-hard monotone clause 3-SAT problem with a polynomial reduction.

A monotone clause 3-SAT problem consists of a set of $n$ variables, and $m$ clauses. Each clause consists of three variables and is either a positive clause or a negative clause. In a positive clause, all three variables appear as literals. In a negative clause, all three variables appear as negated literals. The problem is the same as the standard 3-SAT problem except for the restriction that we cannot mix positive and negative literals in a clause.

We can convert an instance of monotone clause 3-SAT to the binary PO problem as follows. There is only one state where decisions are possible. It is the initial state of the MDP. Each action from this state results in an action-specific probabilistic transition to a set of terminal states, each of which is associated with a terminal reward value.

Because of the simple structure of this MDP, each policy corresponds to an action and vice versa. And, each terminal state corresponds to a reward and vice versa. So, each policy can be viewed as a convex combination of rewards.

We create two terminal states $s_0$ and $s_1$ and create one action ($a_0$) that transitions directly to $s_0$ and one ($a_1$) that transitions directly to $s_1$. We then add a policy constraint that says the

$a_0 < a_1$. Because all rewards are in $\{0, 1\}$, that forces the reward for $s_0$ to be 0 and for $s_1$ to be 1. Those become our logical primitives, in a sense.

Next, we add $2n$ states, one for each positive and negative literal in the 3-SAT problem. For each variable $v$, we add an action $a_v$ with a 50-50 transition to $s_v$ and $s_{\overline{v}}$, along with two constraints: $a_0 < a_v < a_1$. These constraints ensure that the reward assignment to $s_v$ and $s_{\overline{v}}$ can be interpreted as an assignment to the literals where one gets a 1 and the other gets a zero. There is no other way to satisfy these constraints.

Now, for each positive clause $c$ consisting of variables $v_1$, $v_2$, and $v_3$, we create an action $a_c$ that transitions to $s_{v_1}$, $s_{v_2}$, and $s_{v_3}$ with equal probability. We add a policy constraint that $a_c > a_0$, forcing the assignment of rewards to the variables to correspond to a satisfying assignment for that clause. (At least one of the rewards needs to be set to 1.)

For the each negative clause $c$ consisting of variables $\overline{v_1}$, $\overline{v_2}$, and $\overline{v_3}$, we create an action $a_c$ that transitions to $s_{\overline{v_1}}$, $s_{\overline{v_2}}$, and $s_{\overline{v_3}}$ with equal probability. We add a policy constraint that $a_c < a_1$, forcing the assignment of rewards to the variables to correspond to a satisfying assignment for that clause. (At least one of the rewards needs to be set to 0.)

By the way the MDP is constructed, the constraints are satisfied if and only if the rewards represent a satisfying assignments for the given monotone clause 3-SAT formula. Therefore, an efficient solution to the binary PO decision problem would provide an efficient solution to the NP-hard monotone clause 3-SAT problem. □

**Proposition 4.6.** *For any SOAP, PO, or TO, given a finite set of CMPs $\mathcal{E} = \{E_1, \ldots, E_n\}$ with shared state–action space, there exists a polynomial time algorithm that outputs a single reward function that realizes the task (when possible) in each CMP in $\mathcal{E}$.*

***Proof of Proposition 4.6.***

From Theorem 4.3, we know that there is an algorithm to solve the reward design problem for any task and a single environment, in polynomial time. We form the multi-environment algorithm by simply combining the constraints formed by each individual linear program. By the properties of linear programming, the resulting solution will either satisfy *all* of the given constraints, as desired, or will correctly identify that no such satisficing solution exists. □

**Theorem 4.7.** *Task realization is not closed under sets of CMPs with shared state-action space. That is, there exist choices of $\mathcal{T}$ and $\mathcal{E} = \{E_1, \ldots, E_n\}$ such that $\mathcal{T}$ is realizable in each $E_i \in \mathcal{E}$ independently, but there is not a single reward function that realizes $\mathcal{T}$ in all $E_i \in \mathcal{E}$ simultaneously.*

***Proof of Theorem 4.7.***

We consider a pair of CMPs, $(E_X, E_Y)$, with the same three states and two actions. We will show that there exists choice of $\mathcal{T}$ such that $\mathcal{T}$ is realizable in $E_X$ and $E_Y$ independently, but *not* in both simultaneously. Our result assumes we restrict to reward functions that are only a function of state, but we suspect similar cases exist for reward functions on $(s, a)$ pairs and $(s, a, s')$ triples.

Consider the pair of CMPs in Figure B.2. These two CMPs share a state-action space and start-state, but not a transition function (and, say, a $\gamma > 0.5$). Let us further suppose we are interested in the SOAP $\Pi_G = \{\pi_{112}, \pi_{212}\}$, that is, the policies that take $a_1$ in $s_1$, and $a_2$ in $s_2$. In both CMPs, the transition function from $s_0$ transitions to $s_1$ with probability 0.5 and $s_2$ with probability 0.5, for both actions.

We first show that this $\Pi_G$ is realizable in both CMPs. For the CMP on the left, note that the reward function $R(s_1) = 1, R(s_2) = -1$, with $\gamma = 0.95$ will ensure $V^{\pi_{112}}(s_0) = V^{\pi_{212}}(s_0)$, and that both policies are strictly better than all others. Next, note that the same is true for the

example on the right where $R(s_1) = -1, R(s_2) = 1$. Thus, $\Pi_G$ is independently realizable in both of these CMPs.

However, there cannot exist a reward function that makes $\pi_{112}$ and $\pi_{212}$ strictly better than all other policies in both CMPs—it is either better to stay in $s_1$, or to stay in $s_2$, but it cannot be the case that both are true simultaneously. $\qquad\square$

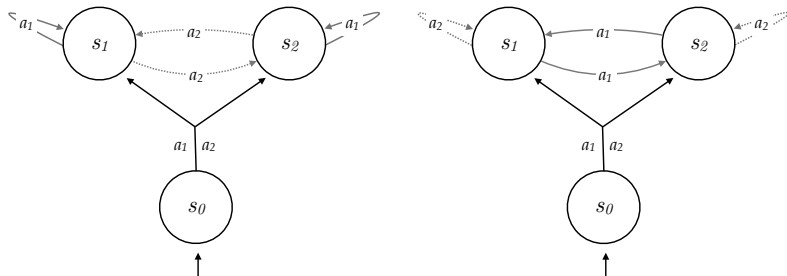

Figure B.2: A pair of CMPs with opposite action effects: On the left, $a_1$ keeps the agent in the same place, while $a_2$ flips the state. On the right, the effects are exactly inverted.

...........................

## C    Experimental Details

Next, we provide further details about our experiments.

### C.1    Expressibility Experiments

First, we provide additional information about the first experiment that explores the fraction of SOAPs that are expressible in small CMPs.

**Six Variants.**    In each figure, we vary one aspect of the environment or task along the x-axis. Most of these are self-explanatory (a: number of actions, b: number of states, c: $\gamma$), though the plots in (d), (e) and (f) are slightly more involved. In (d), we vary the *size* of each sampled SOAP, corresponding to the number of good policies in the SOAP. That is, a SOAP of size one consists only of a single good policy, $\Pi_G = \{\pi^*\}$. In (e), we vary the *Shannon entropy* of the transition function on a per state-action basis as per: $H(T) = -\sum_{s' \in \mathcal{S}} \log_2 T(s' \mid s, a)$. This is accomplished by interpolating between the fully deterministic transition function that only transitions to a single next-state and the uniform random transition function through a simple soft-max distribution in which one next-state is the intended transition, while each other next-state receives a small amount of probability mass depending on the given entropy. In plot (f), we vary the *spread* of the SOAP, which is a measure of how different the good policies in the SOAP are on average. The x-axis corresponds to an approximate edit-distance between policies, where each point on the x-axis defines the parameter of a coin that we flip to determine whether to change a chosen reference policy's action for each state. So, we first randomly sample one policy, say $\pi_1$. Then, we construct the next policy for the SOAP as follows: For a given coin weight $\theta$, we flip a coin at each state of the CMP. If the coin lands heads (the trial is successful), then we change the action of the new policy to a fixed action chosen uniformly at random. Thus, when $\theta$ is zero, the SOAP will only contain $\pi_1$. When $\theta$ approaches one, the SOAP will likely contain many different policies.

**Environment Details.**    In each case, unless otherwise specified, the underlying environment is a four state, three action CMP with $\gamma = 0.95$, and a transition function that is a multinomial over next-states sampled from a Dirichilet-multinomial distribution with $\alpha$ parameters set to $\frac{1}{|\mathcal{S}|}$. When not specified, the size of each SOAP is two. We varied many aspects of these parameters and found little change in the nature of the plots, though trends will be shifted up or down in many cases. For instance, given the downward trend of Figure (3d) as the SOAP size increases, we know that the remaining plots will each be scaled downward if we were to run the same experiment for a SOAP

size larger than two. We sample random SOAPs by first sampling a SOAP size randomly between 1 and $|\Pi|$. Then, we sample $N$ SOAPS of the chosen size uniformly at random (unless otherwise specified, as in the case of Figure (3f)).

## C.2   Learning Experiments

In the grid environment, we set slip probability of 0.35 for all (non-terminal) states. When a "slip" event occurs, the action effect is orthogonal to the intended direction. For instance, in the bottom left cell, if the up action is executed, there is a 0.175 chance the agent will execute `left` (thus staying in the bottom right cell), and a 0.175 chance the agent will execute `right`, and a 0.65 chance the agent will move up a cell. We experiment with tabular Q-learning with $\epsilon$-greedy exploration, with $\epsilon = 0.2$ and learning rate $\alpha = 0.1$ and no annealing. Each episode consists of 10 steps in the environment, with 250 episodes per algorithm. We repeat the experiment 50 times and report 95% confidence intervals. The y-axis measures, at the end of each episode, the (inverse) minimum edit distance between Q-learning's greedy policy and any of the policies in the SOAP along the trajectory taken by Q-learning's greedy policy. Thus, when the series reaches 1.0, Q-learning's greedy policy is identical to one of the SOAP policies in the states that the greedy policy will reach. We observe that the gap between the blue and green curves is due to the different kinds of policies that the SOAP reward and the regular reward promote—one is not necessarily better or worse than the other, they just convey different kinds of objectives.

............................