# OpenReview forum: "On the Expressivity of Markov Reward"
_NeurIPS.cc/2021/Conference — NeurIPS 2021 Oral_

### Official Review · Reviewer_JLWp · 2021-06-26

**Rating:** 8
**Confidence:** 4

**Summary:**

The authors consider the problem of expressing a __task__ via a reward function and discount rate. They show that for various formalizations of what a 'task' is, there exist reward functions which cannot express that task. The authors present an efficient algorithm for deciding the expressibility of a task, and for designing such a reward function. They study certain properties of how many tasks are expressible in a given environment, and demonstrate how their approach can accelerate learning.

**Limitations And Societal Impact:**

No / N/A

**Main Review:**

This paper is well-written, addresses an important problem which has not been formally studied to-date, and presents strong technical results. I found it very clear and well-motivated, and expect it to inspire important future work.

Here are some minor comments and questions:

* 63: prefix 'behavioral policies' with 'deterministic'?
* 70: 'import' -> 'important'
* 144-147: I found the sentence structure ambiguous; it was unclear what ,"no matter which..." was modifying.
* 177-178: identifying reward functions with shared optimal policy sets seems strong. Perhaps weaken the wording? Also, does reward shaping preserve the return-based ordering over policies - not just the optimal policy sets?
* I found "better" SOAP confusing (195-196). Can you give a more concrete example in camera-ready?
* What does it mean for a reward-function family to have infinitely many outputs? (I haven't checked appendix, but this should be clear in the main paper)

POST-REBUTTAL: I thank the authors for their clarifications. My score remains unchanged: clear accept.

**Time Spent Reviewing:**

2.5

---

> ### Author Response · Authors · 2021-08-09
> **Reply to Reviewer JLWp**
>
> We would like to thank the reviewer for their time and thoughtful comments!
>
> We will incorporate the suggestions provided. To respond to each question in turn:
> - Yes, we believe that potential-based shaping should preserve policy ordering, though the main result of Ng/Harada/Russell only shows that potential-based shaping preserves the optimal policy set.
> - Regarding “better” SOAP: We will plan to include a clear & concrete example in the camera ready.
> - On the question, “What does it mean for a reward-function family to have infinitely many outputs?”: This is a great point, and we should definitely include this information in the main text (and we will be sure to include it in the camera ready). We use this phrase to refer to any set of reward functions for which all reward functions in the set have a codomain with cardinality greater than or equal to the natural numbers. We will add text clarifying.

---

### Official Review · Reviewer_Qqv1 · 2021-07-08

**Rating:** 8
**Confidence:** 3

**Summary:**

This paper studies the ability of Markovian reward functions to represent tasks defined as a set of acceptable policies, partial ordering over policies, partial ordering over trajectories. It first provides a negative result showing that some tasks cannot be specified through a Markovian reward in some environments. Then, it designs a set of linear programs to simultaneously check if a specific task can be encoded into a Markovian reward and to compute such reward. The paper includes an empirical evaluation in a simple domain.

**Limitations And Societal Impact:**

The limitations and scope of the work are well-discussed in the text.

**Main Review:**

Post-rebuttal

Having read the rebuttal and other reviews, I am updating my evaluation from 7 to 8.

---

This work is very interesting, it tackles a relevant question on the expressive power of Markovian rewards with clarity and completeness. The main results on the limitation of Markovian rewards fit nicely with the findings of recent works (e.g., Silver et al., Reward is enough, 2021), which conjecture that non-Markovianity is crucial for the expressivity of rewards. The contributions on reward design, binary rewards, and multiple environments look solid and valuable as well, whereas the proofs are carried out with novel arguments.
I do not have any major concern about this paper, and I suggest accepting it.
I provide below some detailed comments.

DETAILED COMMENTS

- I think that the main limitation of this work lies in the answers of the task-definition question (TaskQ, or "What is a task?"), which are somewhat arbitrary. However, the selected task specifications are reasonable, and they are general enough to cover a large portion (if not all) of the objectives that are considered in the RL literature.

- The reward design methods seek for the valid reward function that maximizes epsilon, which takes slightly different meaning in the three formulations, but always acts as a slack variable on the constraints satisfaction.
Can the authors comment on why this specification-robust reward should be preferable? Would it be fundamentally easier to solve the problem for any reward that satisfies the constraints?

- From the informal definition of task provided between lines 162-165 arises quite naturally the dueling cost-minimization and goal-reaching objectives of the stochastic shortest path (SSP) problem formulation (Bertsekas and Tsitsiklis, An analysis of stochastic shortest path problems, 1991).
Do the authors think that the SSP-MDP model could increase the expressivity of Markov reward (cost) w.r.t. the standard MDP model?

- Another interesting direction for future works may study the expressivity of the convex MDP model (see, Zhang et al., Variational policy gradient method for reinforcement learning with general utilities, 2020; Zahavy et al., Reward is enough for convex MDPs, 2021), which allows to specify the objective as any convex function of the expected state visitations rather than a linear combination.

**Time Spent Reviewing:**

6

---

> ### Author Response · Authors · 2021-08-09
> **Reply to Reviewer Qqv1**
>
> We would like to thank the reviewer for their time and thoughtful comments!
>
> We respond to each point in turn:
> - We agree that an important direction for future work is to study more general notions of task.
> - On the note of $\epsilon$ in the LP (“Can the authors comment on why this specification-robust reward should be preferable? Would it be fundamentally easier to solve the problem for any reward that satisfies the constraints?”): This is a really great question. If we do not explicitly maximize epsilon, then we cannot express strict inequalities in constraints of the LP. For instance, if we want to find a reward function that ensures $\pi_1$ is strictly higher in start-state value than $\pi_2$, as is the case for SOAPs, then we need to ensure that there is a non-zero gap between their start-state values. We know there is always a degenerate solution in which all policies have the same value ($R(s,a) = 0, \forall_{s,a}$), and thus maximizing $\epsilon$ ensures we avoid this degenerate case. The reviewer is correct, however, that there is not a strict reason to prefer a value gap of $0.5$ to $0.1$, but the LP does incentivize larger gaps in value. We might speculate that such a gap induces easier learning dynamics, as it is more immediately apparent as to which policy is better than the other. But, this is just speculation, and we believe this matter is a largely empirical fact that depends on the specifics of the environment and learning-algorithm.
> - On SSPs: this is a really interesting point, and one which we did explore for a time as part of this work. We believe that most SSPs can be translated into a PO, but naturally POs require a different kind of knowledge to construct. By this reasoning, however, we suspect that the SSP formulation cannot provide any additional expressivity to Markov rewards since we believe they will have at best the same expressivity as POs. We have not worked through the formal details, however, so it remains an open question (and one that could be of interest for future work).
> - The convex MDP model is quite interesting; thank you for the suggestion. This could also be a fruitful direction for future work.

---

> > ### Comment · Reviewer_Qqv1 · 2021-08-27
> > **Post-rebuttal comment**
> >
> > I would like to thank the authors for their detailed replies. They may consider to mention these other future directions in the conclusive section of the paper.
> >
> > Having read other reviewers' comments and the replies by the authors, I am even more convinced on the value of this paper. Thus, I will update my score from 7 to 8, and I will stick with my acceptance recommendation.

---

### Official Review · Reviewer_8XHY · 2021-07-14

**Rating:** 8
**Confidence:** 3

**Summary:**

- An exploration and a concrete answer to the question of what is a task, and a critical examination of the reward hypothesis. The approach that this paper proposes is to define a set of all possible tasks in a Controlled Markov Process (CMP), and then ask the question, does there exist a Markov reward that realizes the given task.

- The paper provides answers the above question in the negative through a counter-example. In summary, the authors provide three modes of expressing the task intent: set of accetable policies (SOAP), partial ordering over policies (PO), partial ordering over trajectories (TO). The key result is that there are tasks defined as per all three methods that are not realizable as a Markov reward optimization problem

- The authors finally propose a linear-programming based polynomial time reward design algorithm that generates a Markov reward for the SOAP task specification and returns Null if such a reward does not exist.

**Main Review:**

**Overall:** This is an important submission and I strongly recommend acceptance. It tackles a fundamental question about RL, task expressivity and rewards, and proposes an important negative result, namely, even in Markov domains with task specifications defined by a set of acceptable Markov policies, there are certain tasks that cannot be represented through reward maximization paradigm.

**Comments:**

- Difference between 'better' and 'equal-or-better' SOAP definition: I found the definition a bit hard to follow. Specifically, a mathematical definition of  $\epsilon$-optimal would go a long way in clearing this. I believe this distinction might be important for future analysis, considering the gap between the expressible tasks (Figure 3)

- SOAP definition of tasks: A key drawback of this analysis seems to be the way of defining tasks. The SOAP framework necessitates explicit enumeration of all acceptable policies, and often this set is defined implicitly. Therefore the result that given as task and a CMP environment, reward design is a polynomial-time problem is weakened by the need to explicitly enumerate the acceptable policies. Thus the bottleneck very well might be the problem of enumerating acceptable policies and not that of designing a reward that realizes the task

- Relationship between paradigms of task representation: The authors identify non-Markov tasks as out of the scope of this analysis. But I would be very curious to apply this paradigm to Manna and Pnueli's hierarchy of tasks [1], or even to the POMDP tasks.

- Functional Task descriptions: SOAP, PO and TO are very general descriptions of task definitions, but task descriptions from a functional standpoint represent completion of goals or description of path constraints. While the definitions are general enough to capture all possible tasks, the set of 'practical' tasks might be a smaller subset of these, and Markov rewards might represent a larger fraction of these tasks.

I believe this paper adds valuable insight to the discussion on the reward hypothesis, and warrants acceptance. My comments above are reflections on possible extensions of the work, and not a major criticism of the work as presented. I strongly believe that RL research has not concentrated enough on the source and scope of the reward functions in selecting benchmark domains with well known reward functions, and this represents an important initial step.





**Time Spent Reviewing:**

8

---

> ### Author Response · Authors · 2021-08-09
> **Reply to Reviewer 8XHY**
>
> We thank the reviewer for their time and thoughtful comments!
>
> To respond to each point in turn:
> - We will add a cleaner introduction of the two SOAP types, and include a definition of epsilon-optimality to add further clarity (along with an example, as was also suggested by another reviewer).
> - We agree that SOAPs are a limited task type due to their extensional nature (that is, requiring enumeration of good policies). We are interested in performing a similar study on more general task types, and believe this is a great direction for future work. We thank the reviewer for their suggestion to connect to Manna and Pnueli's task hierarchy, and believe this further study on the formal semantics of tasks and their relationship to RL tasks could be a fruitful direction for study.
> - Regarding functional task descriptions; This point resonates with us as well; it is likely useful to carefully isolate the conditions on SOAP/PO/TO that ensure Markov rewards are sufficient (as suggested by another reviewer, too). These particular subsets of SOAPs/POs/TOs might be of interest on their own. We believe this is another useful direction for future work

---

> > ### Comment · Reviewer_8XHY · 2021-08-31
> > **Post-Response**
> >
> > Thank you to the authors for your response. As stated in the review, most of my comments were reflections and possible extensions. I am very happy with the draft, apart from minor changes for presentation and clarifications.

---

### Official Review · Reviewer_jEvD · 2021-07-16

**Rating:** 7
**Confidence:** 4

**Summary:**

In the reinforcement learning context, the paper proposes three possible definitions for the notion of task, as a set of acceptable policies (SOAP), a partial ordering over policies (PO), or a partial ordering over trajectories (TO). For each definition, the authors show that there exists a task and a controlled Markov process for which there is no reward function that can express that task. Furthermore, the authors notably provide a polynomial algorithm based on linear programming to find such reward function if it exists. Some experiments on random problem instances are also performed to validate the theory.

**Limitations And Societal Impact:**

No discussion is needed since this work is more fundamental.

**Main Review:**

The paper is well-written and easy to understand. Although I find the theoretical results quite simple, I like very much the whole formalization that provides a clear framework for describing which situations Markov rewards could be sufficient or not to express some tasks. I think this provides a good initial formal answer to the vaguely formulated reward hypothesis.

The paper could have been strengthened by also considering conditions in SOAP, PO, or TO under which Markov rewards are sufficient. However, I believe this initial work could lead to further work, which could either refine the current results or consider more sophisticated definitions of tasks.

Minor comments:
- line 70: import
- line 225: then
- line 273: I think this sentence doesn't apply to PO.
- line 274: polices
- line 342: the there

**Time Spent Reviewing:**

4

---

> ### Author Response · Authors · 2021-08-09
> **Reply to Reviewer jEvD**
>
> We thank the reviewer for their thoughtful comments! Your suggestions (labeled "Minor comments:") will be incorporated into the camera ready. Additionally, we agree with the suggestion to think carefully about identifying the conditions on SOAP/PO/TO that ensure Markov rewards are sufficient, and believe this is a great direction for further study.

---

### Decision · Program_Chairs · 2021-09-27

**Decision:**

Accept (Oral)

**Comment:**

Reviewers agree that this paper is interesting, relevant, novel, clear, well-written, and technically sound.
I congratulate the authors for their work and I invite them to modify their paper following the reviewers' suggestions.